# CT scan pancreatic cancer segmentation and classification using deep learning and the tunicate swarm algorithm

**Hari Prasad Gandikota**[1]*, **Abirami S.**[1], **Sunil Kumar M.**[2]

**1** Department of Computer Science & Engineering, Annamalai University, Chidambaram, Tamilnadu, India, **2** School of Computing, Mohan Babu University, Tirupati, Andhra Pradesh, India

* csedept.au@gmail.com

**Data Availability Statement:** All relevant data are within the paper.

**Funding:** The authors received no specific funding for this work.

## Abstract

Pancreatic cancer (PC) is a very lethal disease with a low survival rate, making timely and accurate diagnoses critical for successful treatment. PC classification in computed tomography (CT) scans is a vital task that aims to accurately discriminate between tumorous and non-tumorous pancreatic tissues. CT images provide detailed cross-sectional images of the pancreas, which allows oncologists and radiologists to analyse the characteristics and morphology of the tissue. Machine learning (ML) approaches, together with deep learning (DL) algorithms, are commonly explored to improve and automate the performance of PC classification in CT scans. DL algorithms, particularly convolutional neural networks (CNNs), are broadly utilized for medical image analysis tasks, involving segmentation and classification. This study explores the design of a tunicate swarm algorithm with deep learning-based pancreatic cancer segmentation and classification (TSADL-PCSC) technique on CT scans. The purpose of the TSADL-PCSC technique is to design an effectual and accurate model to improve the diagnostic performance of PC. To accomplish this, the TSADL-PCSC technique employs a W-Net segmentation approach to define the affected region on the CT scans. In addition, the TSADL-PCSC technique utilizes the GhostNet feature extractor to create a group of feature vectors. For PC classification, the deep echo state network (DESN) model is applied in this study. Finally, the hyperparameter tuning of the DESN approach occurs utilizing the TSA which assists in attaining improved classification performance. The experimental outcome of the TSADL-PCSC method was tested on a benchmark CT scan database. The obtained outcomes highlighted the significance of the TSADL-PCSC technique over other approaches to PC classification.

## 1. Introduction

Currently, pancreatic cancer (PC) is the most incurable and lethal disease of which survival rates have not yet been greater significantly [1]. Magnetic Resonance Imaging (MRI) guided radiotherapy is nowadays used to decrease cancer; but anatomical changes, i.e. breathing does not affect it due to the variability and interpatient infarction. Initial and precise detection of the PC is a challenge [2]. Enhancing earlier detection, early treatment, and initial diagnosis is

**Competing interests:** The authors have declared that no competing interests exist.

of utmost importance. Computer-aided diagnoses (CAD) system has been devised with the advancements of computer science and image processing technologies for disease diagnoses [3]. Radiotherapists commonly use CAD systems to enhance diagnostic accuracy, help in detecting and interpreting disease, and decrease the burden on physicians. CAD method was recently developed in deep neural networks (DNNs) and prolonged the need for health care services [4]. Higher pathology in PC resulted in significant interest in optimizing effectual treatments and CAD systems where accurate pancreatic segmentation was required. Hence, there comes a need to develop a new approach for pancreatic segmentation. Today, computed tomography (CT) segmentations of the pancreas become a challenge. The most significant element of the CAD is image recognition [5]. The process of detecting adenocarcinomas has 2 stages: feature extraction and feature selection.

Current advancements in deep learning (DL) have witnessed greater potentiality in medical image analysis [6]. In the earlier study, it is proved that a convolutional neural network (CNN) can precisely differentiate between PC and noncancerous pancreas. But the radiologists manually perform identification of the pancreas with the help of CNN [7]. Identification of segmentation of the pancreas becomes challenging as the pancreas borders many structures and organs and differs in size and shape, particularly in patients with PC. Still, a medically applicable CAD tool must enable classification and segmentation (i.e., forecasting the absence or presence of PC), with minimum labour or human annotation [8]. The DL approach utilizing CNN has proved much more potential in examining clinical images. The neural network (NN) construction based on neurons contains activation parameters and functions that extract and merge features in the images and establishes a method that captured intricate relationships between images and diagnoses [9]. In the imaging identification of conditions like skin tumor, diabetic retinopathy (DR), and liver masses, CNN achieves greater performance. Still, the potential CNN advantages for diagnosing PC have not been studied widely [10]. Typically, PC is unclear at an initial stage imposes issues for trained radiotherapists and presents with ill-defined margins and irregular contours on CT.

The study developed the tunicate swarm algorithm with deep learning-based pancreatic cancer segmentation and classification (TSADL-PCSC) technique on CT scans. The TSADL-PCSC technique aims to accomplish enhanced PC classification results using a hyperparameter-tuned DL model. Primarily, the TSADL-PCSC technique employs a W-Net segmentation approach to define the affected regions on the CT scans. Besides the TSADL-PCSC technique utilizes GhostNet feature extractor for generating a group of feature vectors. For PC classification, the deep echo state network (DESN) model is applied in this study. At last, the hyperparameter tuning of the DESN approach occurs utilizing the TSA which assists in attaining improved classification performance. The simulation results of the TSADL-PCSC algorithm are tested on a benchmark CT scan dataset.

## 2. Related works

Vaiyapuri et al. [11] present an IDLDMS-PTC (intelligent DL-assisted decision-making medical system for PC classification) model with CT scans. The proposed algorithm develops an emperor penguin optimizer (EPO) using the multi-level thresholding (EPO-MLT) method for the segmenting PC. Moreover, the MobileNet architecture was employed as a feature extraction with optimum autoencoder (AE) for the classification of PC. The authors in [12] present and validate a DL architecture, which integrates level-set, and multi-atlas registration for the segmentation of PC from CT scans. The presented algorithm comprises three phases such as refine, coarse, and fine phases. Initially, by using multi-atlas-based 3D diffeomorphic registration and fusion, a coarse segmentation can be attained. Next, three 2-D slice-related CNNs

and 3-D patch-related CNN were utilized for the prediction of a fine segmentation. For the completely automated predictive of preoperative pathological grading of PC, Zhang et al [13] introduced a DL algorithm in this study. A DL approach for the PC segmentation was coined first to attain lesion region. Next, each patient was divided into a test set, training set, and validation set. The features calculated from the lesion region introduced a prediction method of PC pathological grade. Lastly, the model stability was confirmed by seven-fold cross-validation.

The authors in [14], designed an ODL-PTNTC (optimum DL-based PC and non-tumour classification) algorithm using CT images. This presented method exploits the adaptive window filtering (AWF) method for noise removal. Furthermore, the sailfish optimizer-based Kapur's Thresholding (SFO-KT) method was used for the process of segmentation. Besides, Political Optimizer (PO) with Cascade Forward NN (CFNN) was used for the classifier purpose. Bagheri et al. [15] utilized a deep CNN (DCNN) for pancreas segmentation in an openly accessible dataset. By using the Dice similarity coefficient (DSC), the accuracy of the segmentations was evaluated. Khdhir et al. [16] developed an ALO-CNN-GRU mechanism for the segmentation and classification of PC depending on DL and CT images. The images undergo pre-processing for noise reduction. The segmentation was processed by the Antlion Optimization (ALO) technique. The segmentation can be performed by using the classifier of the CNN and Gated Recurrent Unit (GRU) models.

Nishio et al. [17] introduced and evaluated a combination of DL architectures and data augmentation methods for automated pancreas segmentation on CT scans. Deep U-Net and Baseline U-Net are selected for the DL algorithms of pancreas segmentation. Data augmentation techniques involved random image cropping and patching (RICAP), mixup, and conventional method. Yang et al. [18] introduced AX-Unet, a DL architecture integrating an improved atrous spatial pyramid pooling model for learning the location data and for extracting multi-level contextual data for reducing data loss in the course of downsampling. Also, a group convolution model was introduced on the feature map at all the levels for achieving data decoupling between channels. Moreover, an explicit boundary-aware loss function was proposed for tackling blurry boundary problems. Compared to radiologist interpretation, the authors in [19], investigated whether CNN discriminates individuals with and without PC on CT. Images are pre-processed into patches, and a CNN was trained for the classification of patches as tumorous or non-tumorous.

## 3. The proposed model

In this manuscript, we have developed the TSADL-PCSC method for PC segmentation and classification on CT scans. The purpose of the TSADL-PCSC technique is to design an effectual and accurate model to improve the diagnostic performance of PC. To accomplish this, the TSADL-PCSC technique comprises four processes namely W-Net segmentation, GhostNet feature extractors, DESN classification, and TSA-based hyperparameter tuning. Fig 1 describes the working flow of the TSADL-PCSC system.

### 3.1. Image segmentation: W-Net model

At the initial stage, the input CT scans are passed into the W-Net model for the segmentation process. The W-net-based segmentation network has been used to attain the segmentation map of the CT scans [20]. Using the decoding and encoding path, this model preserves the localization and content information. Furthermore, edge data maintain consistency and are preserved to sharpen the image during segmentation. This network was planned as a progression of *U*-Net. Later, by connecting both *U*-Net topologies, a single AE was implemented. In

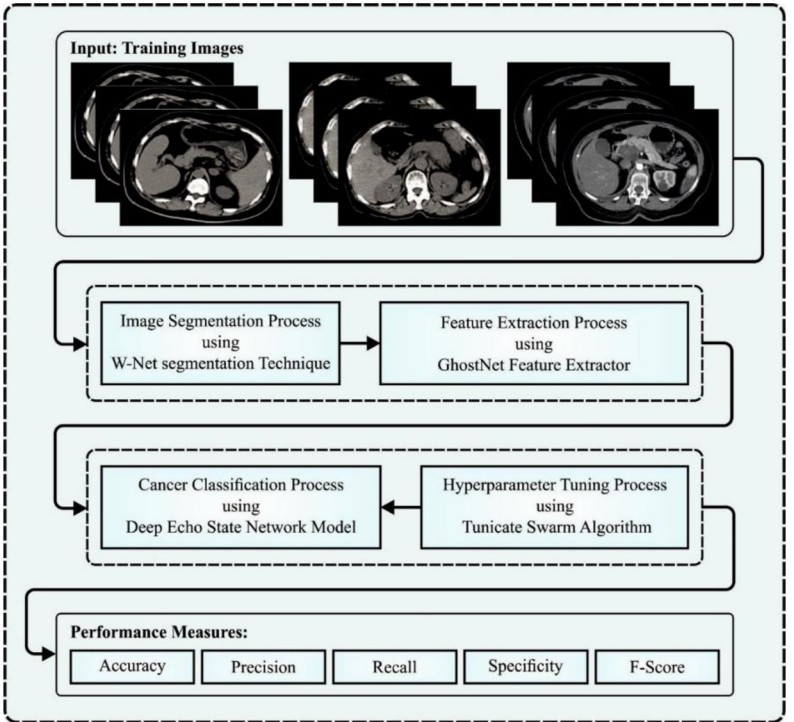

**Fig 1. Working flow of TSADL-PCSC approach.**

*U*-Net, an encoder (contracting path) and a decoder (expansive path) based architecture were applied.

The 1st module of *the W*-net was the encoder that encompassed a set of blocks. The three-layer convolution and BN layers interspersed with ReLU was the essential component of the block. The basic module is considered twice for creating a single convolutional block. The blocks were joined by using 2×2 layers of max pooling. Using max pooling, the count of parameters is reduced and the critical target data was preserved. During the decoder, the kernel count of a convolutional layer is 8, increasing from 8 to 128 during the encoder.

The second-wide path was the expansive path. Convolution and Upsampling layers made up its infrastructure. During the contracting path, the input has been downscaled just once, and in the expansive path, the input was upscaled four times. The mapping feature from the contracting path was concatenated with corresponding mapping features in the expansive path for recovering lost data in max pooling and convolution procedures. The second part is corresponding to the first, however, the outcome of the top pooling layer was integrated, and the outcome of the unit was placed at the same level in the first *U*-Net.

Like others, there is an additional block which follows the upsampling of the contracting path and the last combination of the expansive initial block. At last, a 1×1 convolutional layer and a softmax activation function were used for matching the desirable amount of classes and mapping features. The cross-entropy loss (CEL) and total-variation loss (CT-loss) are

combined in this method.

$$Loss = L_{cr-etrp} + L_{total-var} \tag{1}$$

$$L_{cr-etrp} = L(sr'_n, pc_n) = -\sum_i^K pc_i \log(sr'_i) \tag{2}$$

$$L_{total-var} = L\{sr\})'_n = \sum_\xi^{W-1} \sum_\eta^{H-1} \|sr'_{\xi+1,\eta} - sr'_{\xi,\eta}\| \|sr'_{\xi+1,\eta} - sr'_{\xi,\eta}\|1 \tag{3}$$

In the formula, *W* and *H* characterize the width and height of input images, correspondingly. $\left\{ sr'_{n;} \right\}$ refers to the sample *n*'s normalized segmentation maps; {*PC*} represents the pseudo segmentation mask made by the index which increases the segmentation map values. This CT loss assists in reducing the time and memory used. Also, the segmentation mask was significantly compressed, which negates the need for post-processing due to the features of the CT loss.

### 3.2. Feature extraction: GhostNet model

To derive a set of feature vectors, the GhostNet model is used. The GhostNet model removes features with some parameters and efficiently receives unwanted data from the network [21]. The GhostNet element turns the typical convolutional function into 2-step operations. A primary stage is the typical convolutional function, however, it decreases the application of the convolutional kernels. The secondary stage is a lightweight linear function for generating redundant mapping features. Once the dimensional of the input mapping feature is $D_F x D_F x M$, the convolutional kernel of standard convolutional is $D_k x D_k x N$, and the calculation amount is $D_k x D_k x M x D_F x D_F x N$. A primary stage of the GhostConv element considers that *m* mapping features can be created, and the calculation amount is $D_k x D_k x M x D_F x D_F x m$. For ensuring a similar size as the typical convolutional output, the secondary stage of the GhostConv element was a lightweight linear function on mapping feature outcome by the primary stage, as depicted in Eq (4).

$$y_{ij} = \phi_{ij}(y'_i), \forall_i = 1, \cdots, m; j = 1, \cdots, s, \tag{4}$$

whereas $\phi_{ij}$ represents a linear operation, $y'_i$ implies the $i^{th}$ mapping feature, *and* $y_{ij}$ stands for the $j^{th}$ mapping feature attained by the linear operation of the $i^{th}$ mapping feature. The Ghost-Conv model gets *N* output mapping features, and *N = mxs*. It has been demonstrated that *s*-1 linear conversion that deals with computational resources was carried out, therefore the computing count of the GhostConv model can be $D_K x D_K x M x D_F x D_F x m + (s-1) x D_K x D_K x D_F x D_F$. Then the computation relationship among GhostConv and typical convolutional modules is expressed as follows

$$\frac{D_K x D_K x M x D_F x D_F x N}{D_K x D_K x M x D_F x D_F x m + (s-1) x D_K x D_K x D_F x D_F} \approx s, \tag{5}$$

According to Eq (5), the typical convolutional is *s* times as much as the GhostConv element in the computation. Thus, the GhostNet model was built depending on GhostNet-Block could considerably decrease the count of computation and the count of network parameters.

### 3.3. Image classification: DESN model

In this work, the DESN approach can be employed for the detection and classification of PC. The important feature of the ESN is that it proceeds a random reservoir as a fundamental

processing unit [22]. The reservoir was stimulated as a difficult internal state, which describes the feature of an input signal by the linear incorporation. Furthermore, the input weights and reservoir were fixed, and just the output weight was modified by linear regression during training of the ESN, which could avoid local minima, exploding, and vanishing gradients and improve efficiency. Fig 2 displays the infrastructure of ESN.

Consider = $\{x_1, x_2, \cdots, x_{N-1}, x_N\}$ represent the internal state of reservoirs, $u = \{u_1, u_2, \cdots, u_{n-1}, u_n\}$ as the input signals, $x$ and $y = \{y_1, y_2, \cdots, y_{m-1}, y_m\}$ denotes the output signals:

$$x(t+1) = f(W_{in}u(t+1) + Wx(t) + W_{back}y(t)) \qquad (6)$$

In Eq (6), $f(\cdot)$ denotes an activation function and $W_{in}$, $W$, and $W_{back}$ are random input, internal, and feedback weights, correspondingly. The leaky combine was considered as a neuron once the ESN was utilized for pattern detection, hence Eq (1) is changed as:

$$x(t+1) = (1 - \alpha\gamma)x(t) + \gamma f(W_{in}u(t+1) + Wx(t) + W_{back}y(t)) \qquad (7)$$

In Eq (7), $\alpha$ denotes the leaky rate and $\gamma$ refers to the gain:

$$x(t+1) = (1 - \alpha)x(t) + f(W_{in}u(t+1) + Wx(t)) \qquad (8)$$

The output of ESN is:

$$y(t) = g(W_{out}[u(t); x(t)]) \qquad (9)$$

Where $g(\cdot)$ refers to the activation function and $W_{out}$ denotes the output weight.

$W_{out}$ is updated during the training of the ESN. The objective function $L$ can be represented as follows:

$$L(\widehat{W}_{out}) = \|g^{-1}(y) - W_{out}[u; x]\|_2^2 \qquad (10)$$

In Eq (10), $\|\cdot\|_2$ refers to $L_2$ norm and $g^{-1}(\cdot)$ shows the inverse function of $g(\cdot)$.

The resultant weighted $\widehat{W}_{out}$ predicted was:

$$\widehat{W}_{out} = g^{-1}(y)[u; x]^{\dagger} = g^{-1}(y)([u; x]^T[u; x])^{-1}[u; x]^T \qquad (11)$$

In Eq (11), the pseudo-inverse and the transpose of the matrix can be represented as superscripts $\dagger$ and $T$, correspondingly.

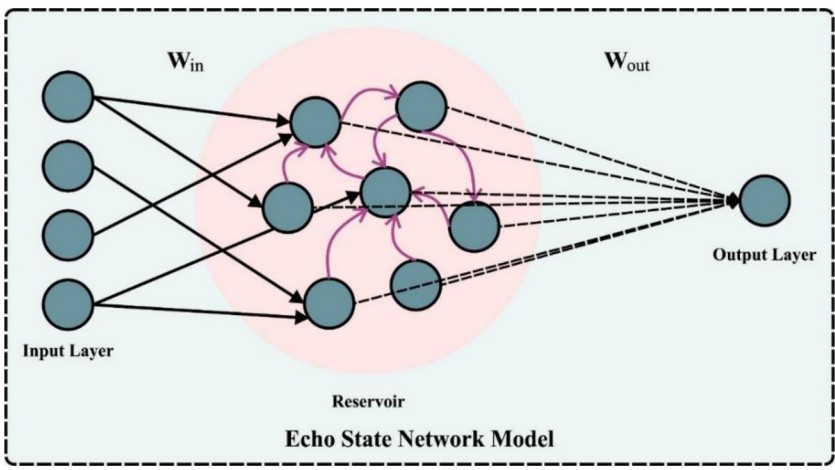

**Fig 2. Architecture of ESN.**

### 3.4. Hyperparameter tuning: TSA model

For optimum hyperparameter tuning of the DESN model, the TSA is used. TSA is inspired by the social behavior of tunicates watching for prey [23]. While hunting, the marine invertebrate exploits swarm intelligence and water jet to find prey. All the tunicates could quickly release the inhaled seawater using siphons of the atrium that make a type of jet propulsions which drives it quickly. Besides, tunicate displayed SI once it can share search details about the food location. The tunicate must meet the succeeding three limitations for establishing the mathematical modelling of its jet propulsion method:

- Avoid clashes among all the search agents.

- All the agents are guaranteed to move towards the fittest individual.

- Create the search agent joins toward the region adjacent to the fittest individual.

To prevent clashes between each search agent, the below formula was used for calculating the novel location of the agent:

$$\vec{A} = \frac{\vec{G}}{\vec{M}} \tag{12}$$

$$\vec{G} = c_2 + c_3 - \vec{F} \tag{13}$$

$$\vec{F} = 2 \cdot c_1 \tag{14}$$

Where $\vec{A}$ denotes the vector used to find the newest location of all the agents; $\vec{G}$ refers to gravity; $F$ indicates water flow in the deep sea; and $c_1$, $c_2$, and $c_3$ represent three random integers within [0,1]. $\vec{M}$ denotes the vector value as social strength between the searching agents as follows:

$$\vec{M} = P_{\min} + c_1 \cdot (P_{\max} - P_{\min}) \tag{15}$$

In Eq (15), $P_{\min}$ and $P_{\max}$ signify the primary and secondary speeds that allow the search agent to construct social interaction and $P_{\min}$ and $P_{\max}$ are fixed to 1 and 4.

Each one move towards the neighbouring individual with the maximal fitness values (FV), after solving clashes between neighbouring search agents as follows:

$$\vec{PD} = \left| \vec{X_{best}} - r_{rand} \cdot \vec{X(t)} \right| \tag{16}$$

In Eq (16), $\vec{X_{best}}$ denotes the food at the position of the present optimum individual; $\vec{PD}$ denotes a vector, which is the spatial distance among target food as well as tunicates; $r_{rand}$ denotes the arbitrary integer in zero and one; and $\vec{X(t)}$ shows the location data of the present search agent at $t$-$th$ iterations.

To create the search agent and execute sufficient local exploration of neighbouring fittest individuals for finding the better solution of the present iteration, the location was evaluated by:

$$X(t) = \begin{cases} X_{best} - \vec{A} \cdot \vec{PD}, \ if \ r_{rand} < 0.5 \\ X_{best} + \vec{A} \cdot \vec{PD}, \ if \ r_{rand} \geq 0.5 \end{cases} \tag{17}$$

At $t$ iteration, all the searching agents explore the region adjacent to the fittest individual $X_{best}$ and allocate the outcome to $X(t)$ for upgrading the place.

The swarming behaviour of the tunicates transfers location data between the searching agents. This process can be driven by the location of the present search agents and can be attained as per the location upgraded by the present search agents. The fittest individual and the upgraded place by the prior individual using the swarm act can accomplish this:

$$X_i\left(\vec{t}+1\right) = \rightarrow \begin{cases} \dfrac{X_i(\vec{t}) + X_{i-1}(\vec{t}+1)}{2 + c_1} & if\ i > 1 \\ X_i(\vec{t}) & if\ i = 1 \end{cases} \tag{18}$$

Here $i = 1, \ldots, N$, $N$ denotes the population size, $X_i\left(\vec{t}+1\right)$ shows the location of the existing search agents, $and\ X_{i-1}(\vec{t}+1)$ represent the place of prior search agents at the following iteration.

To demonstrate the procedure of TSA, the steps to upgrade the location of the search agent are given below:

Step1: Initializes the original population of searching agents $X$.

Step2: Allocate value to initial parameters and max -iterations.

Step3: Evaluate the FV of all the tunicates and choose the individual with better FV as a better search agent.

Step4: Upgrade the place of all the search agents based on Eq (18).

Step5: Keep all the search agents from the search space.

Step6: Measure the FV of all the upgrade searching agents; if there were fittest individuals than the prior best-searching agents from the population, update $X_{best}$.

Step7: If the maximum iteration was obtained, and the process stops. Or else, return to steps 4 to 7.

Step8: Print the optimum individual ($X_{best}$)

The TSA system produces a fitness function (FF) to obtain greater efficacy of classification. It defines positive integers to signify the better outcome of the solution candidate. The decline of the classifier error rate is regarded as FF.

$$fitness(x_i) = ClassifierErrorRate(x_i) = \frac{number\ of\ misclassified\ samples}{Total\ number\ of\ samples} * 100 \tag{19}$$

## 4. Performance validation

The pancreatic cancer classification results of the TSADL-PCSC method are tested on the benchmark BioGPS datasets [3]. The dataset consists of 500 samples with two classes [24] as represented in Table 1.

**Table 1. Details of database.**

| Class | No. of Samples |
|---|---|
| Pancreatic Tumor | 250 |
| Non-Pancreatic Tumor | 250 |
| Total Samples | 500 |

In Fig 3, the confusion matrix of the TSADL-PCSC technique is analysed on pancreatic cancer classification. The results indicate that the TSADL-PCSC technique recognized pancreatic cancer and non-pancreatic cancer proficiently.

In Table 2 and Fig 4, the PC classifier result of the TSADL-PCSC method under 80:20 of TRP/TSP. The experimental values detect pancreatic cancer proficiently. For example, on 80% of TRP, the TSADL-PCSC method gains average $accu_y$ of 96.98%, $prec_n$ of 97.18%, $sens_y$ of 96.98%, $spec_y$ of 96.98%, and $F_{score}$ of 97%. At the same time, on 20% of TSP, the TSADL-PCSC technique gains average $accu_y$ of 99.02%, $prec_n$ of 99%, $sens_y$ of 99.02%, $spec_y$ of 99.02%, and $F_{score}$ of 99%.

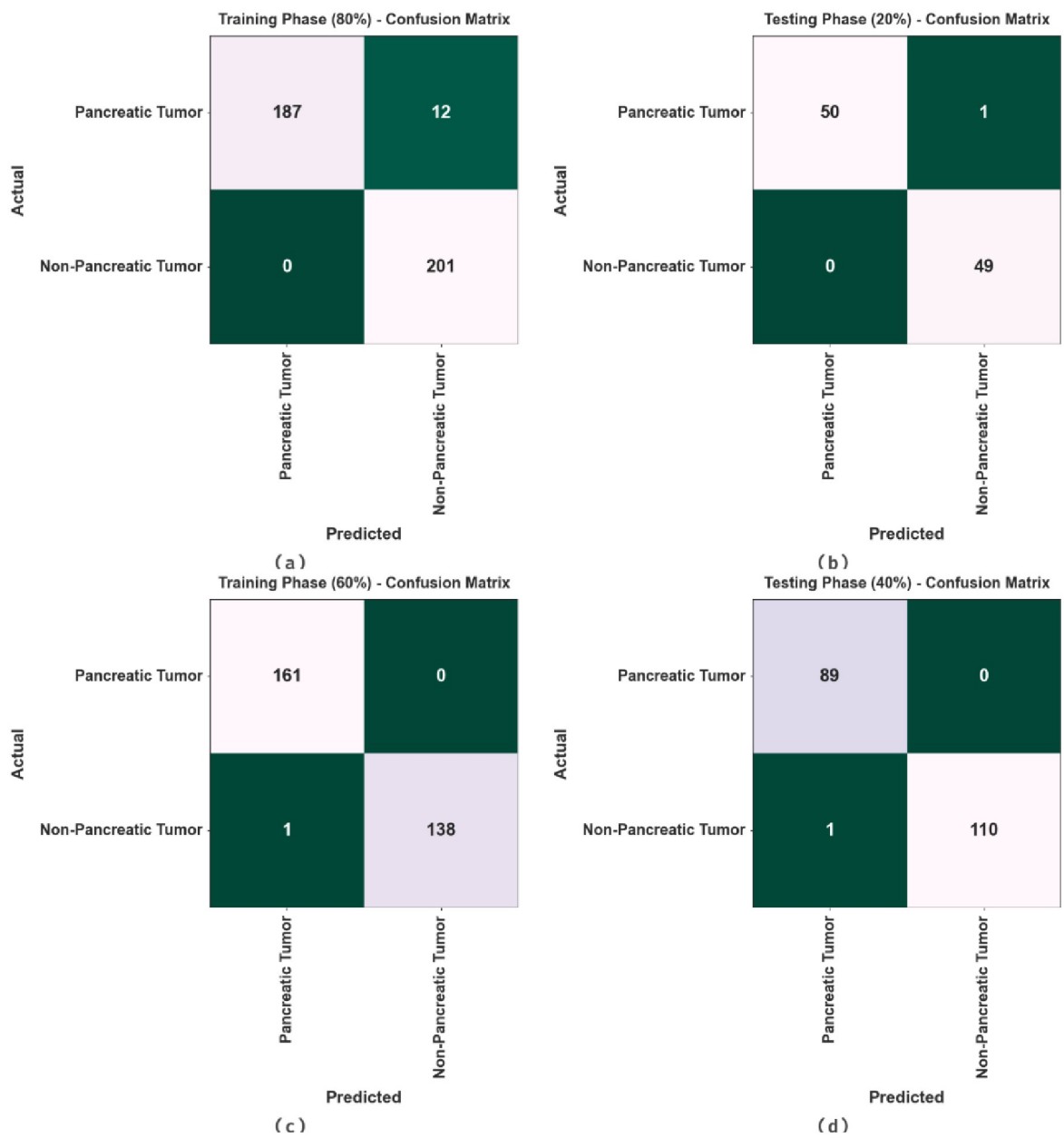

**Fig 3. Confusion matrices of TSADL-PCSC method (a-b) 80:20 of TRP/TSP and (c-d) 60:40 of TRP/TSP.**

**Table 2. PC classifier outcome of TSADL-PCSC technique on 80:20 of TRP/TSP.**

| Class | $Accu_y$ | $Prec_y$ | $Sens_y$ | $Spec_y$ | $F_{score}$ |
|---|---|---|---|---|---|
| Training Phase (80%) | | | | | |
| Pancreatic Tumor | 93.97 | 100.00 | 93.97 | 100.00 | 96.89 |
| Non-Pancreatic Tumor | 100.00 | 94.37 | 100.00 | 93.97 | 97.10 |
| Average | 96.98 | 97.18 | 96.98 | 96.98 | 97.00 |
| Testing Phase (20%) | | | | | |
| Pancreatic Tumor | 98.04 | 100.00 | 98.04 | 100.00 | 99.01 |
| Non-Pancreatic Tumor | 100.00 | 98.00 | 100.00 | 98.04 | 98.99 |
| Average | 99.02 | 99.00 | 99.02 | 99.02 | 99.00 |

In Table 3 and Fig 5, the PC classifier outcome of the TSADL-PCSC system under 60:40 of TRP/TSP. The experimental values detect pancreatic cancer proficiently. For instance, on 60% of TRP, the TSADL-PCSC technique gains an average $accu_y$ of 99.64%, $prec_n$ of 99.69%, $sens_y$ of 99.64%, $spec_y$ of 99.64%, and $F_{score}$ of 99.66%. Simultaneously, on 40% of TSP, the TSADL-PCSC method gains an average $accu_y$ of 99.55%, $prec_n$ of 99.44%, $sens_y$ of 99.55%, $spec_y$ of 99.55%, and $F_{score}$ of 99.49%.

Fig 6 examines the $accu_y$ of the TSADL-PCSC method during training and validation processes on 60:40 of TRP/TSP. The figure notifies that the TSADL-PCSC technique attains the highest $accu_y$ values over maximum epochs. Furthermore, the maximum validation $accu_y$ over training $accu_y$ exhibits that the TSADL-PCSC methodology attains effectively at 60:40 of TRP/TSP.

The loss analysis of the TSADL-PCSC method during training and validation is illustrated on 60:40 of TRP/TSP in Fig 7. The outcome indicates that the TSADL-PCSC method attains

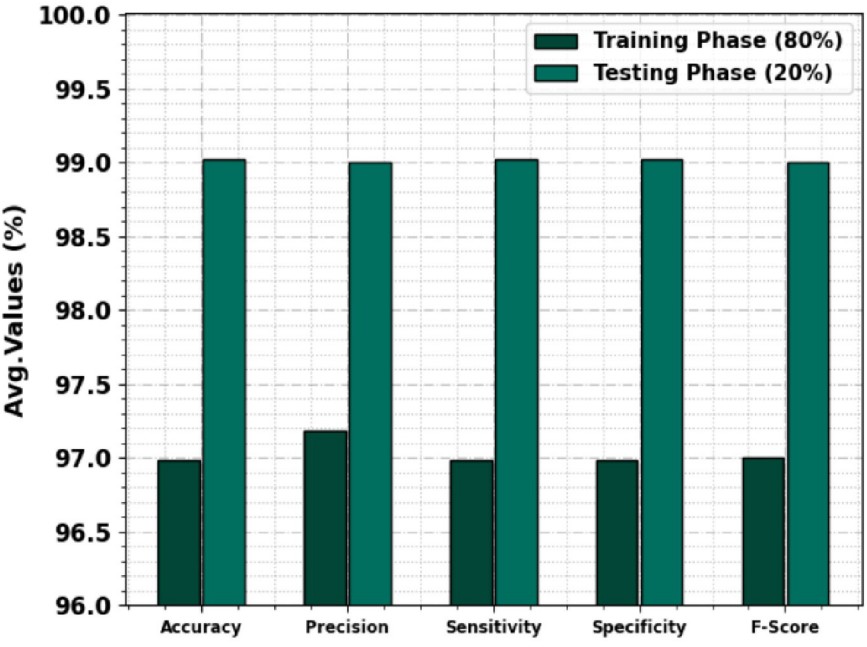

**Fig 4. Average outcome of TSADL-PCSC approach on 80:20 of TRP/TSP.**

**Table 3. PC classifier outcome of TSADL-PCSC technique on 80:20 of TRP/TSP.**

| Class | $Accu_y$ | $Prec_y$ | $Sens_y$ | $Spec_y$ | $F_{score}$ |
|---|---|---|---|---|---|
| Training Phase (60%) | | | | | |
| Pancreatic Tumor | 100.00 | 99.38 | 100.00 | 99.28 | 99.69 |
| Non-Pancreatic Tumor | 99.28 | 100.00 | 99.28 | 100.00 | 99.64 |
| Average | 99.64 | 99.69 | 99.64 | 99.64 | 99.66 |
| Testing Phase (40%) | | | | | |
| Pancreatic Tumor | 100.00 | 98.89 | 100.00 | 99.10 | 99.44 |
| Non-Pancreatic Tumor | 99.10 | 100.00 | 99.10 | 100.00 | 99.55 |
| Average | 99.55 | 99.44 | 99.55 | 99.55 | 99.49 |

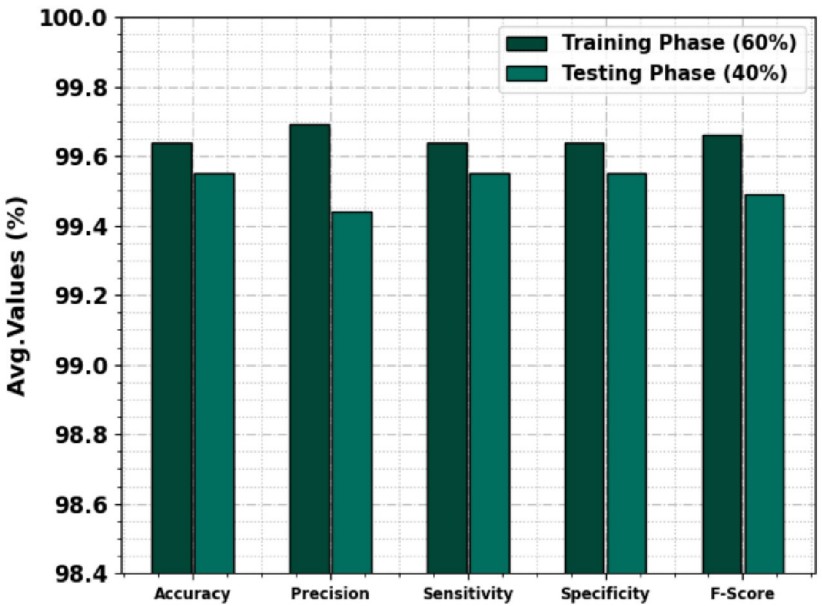

**Fig 5. Average outcome of TSADL-PCSC technique on 70:30 of TRP/TSP.**

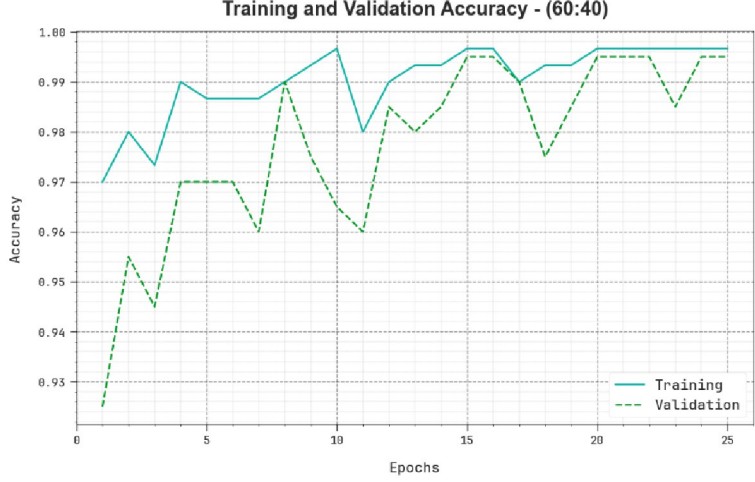

**Fig 6. Accuracy curve of TSADL-PCSC approach on 60:40 of TRP/TSP.**

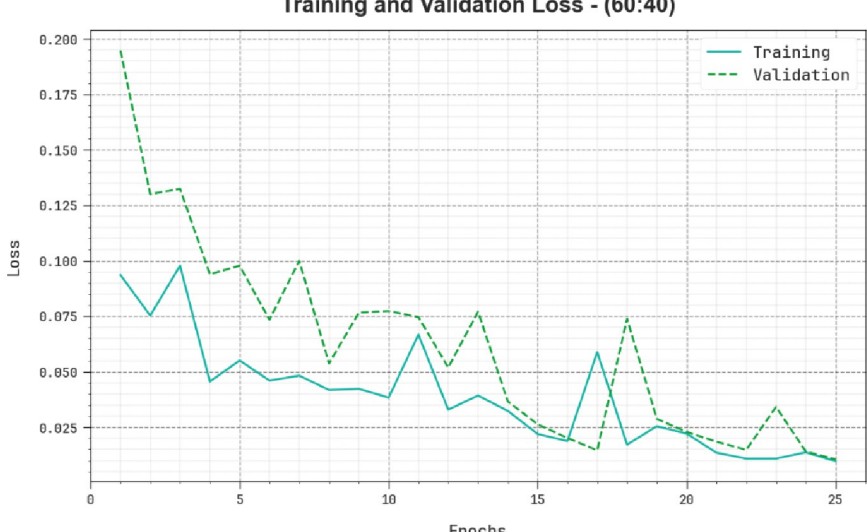

**Fig 7. Loss curve of TSADL-PCSC approach on 60:40 of TRP/TSP.**

nearby values of training and validation losses. The TSADL-PCSC method efficiently gains on 60:40 of TRP/TSP.

A brief precision-recall (PR) analysis of the TSADL-PCSC technique is shown on 60:40 of TRP/TSP in Fig 8. The outcome stated that the TSADL-PCSC approach outcomes in the highest values of PR. The TSADL-PCSC method could obtain the highest PR values in 2 classes.

In Fig 9, a ROC analysis of the TSADL-PCSC technique is shown on 60:40 of TRP/TSP. The figure defines that the TSADL-PCSC method resulted in maximum ROC values. In addition, the TSADL-PCSC system shows maximum ROC values on all classes.

A brief comparison study is made in Table 4 and Fig 10 in order to highlight the outperforming outcomes of the TSADL-PCSC method [11]. The outcome indicates that the CNN-

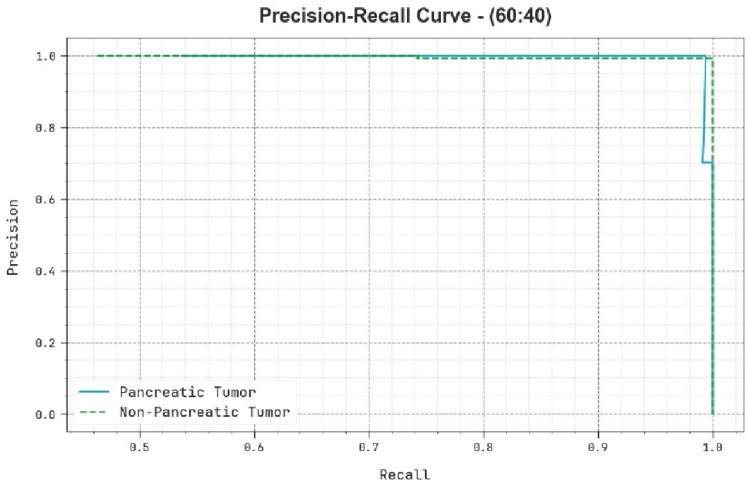

**Fig 8. PR curve of TSADL-PCSC approach on 60:40 of TRP/TSP.**

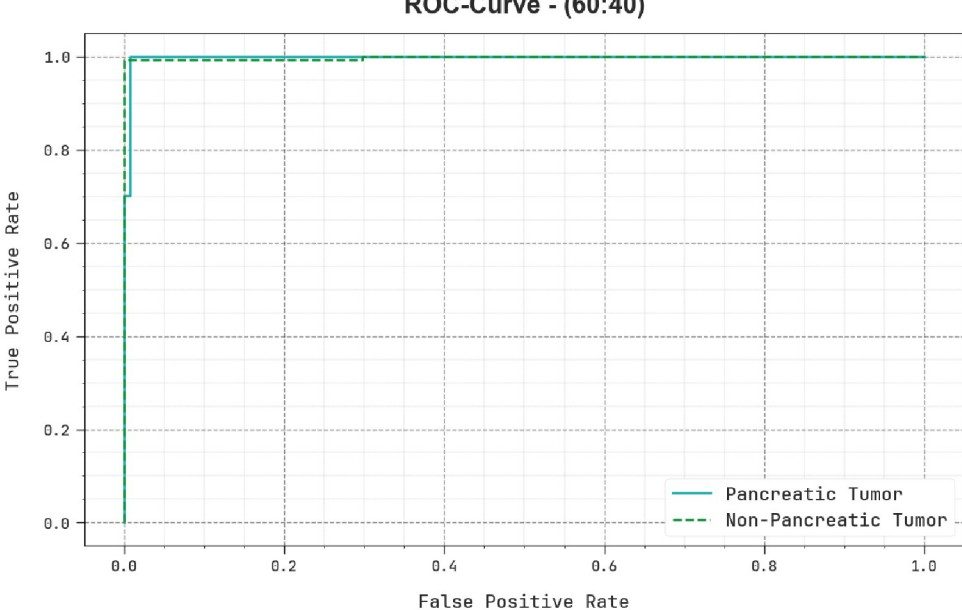

**Fig 9. ROC curve of TSADL-PCSC approach on 60:40 of TRP/TSP.**

50x50 model has obtained poor performance with the least results. In addition, the ODL-PTNTC, WELM, KELM, and ELM algorithms have attained slightly improved performance. Although the IDLDMS-PTC technique reaches near-optimal performance, the TSADL-PCSC technique gains outperforming results with maximum $sens_y$ of 99.55%, $spec_y$ of 99.55%, and $accu_y$ of 99.55%. These results indicate the promising performance of the TSADL-PCSC technique in terms of different measures.

## 5. Conclusion

In this study, we have developed the TSADL-PCSC method for PC segmentation and classification on CT scans. The TSADL-PCSC technique aims to accomplish enhanced PC classification results using a hyperparameter-tuned DL model. To accomplish this, the TSADL-PCSC technique comprises four processes namely W-Net segmentation, GhostNet feature extractor, DESN classification, and TSA-based hyperparameter tuning. The TSA helps to avoid the manual trial and error hyperparameter selection process, which in turn increases the overall classification performance. The experimental result of the TSADL-PCSC method was tested on a benchmark CT scan database. The obtained outcomes highlighted the importance of the

**Table 4. Comparative outcome of TSADL-PCSC method with other techniques.**

| Methods | Sensitivity | Specificity | Accuracy |
|---|---|---|---|
| TSADL-PCSC | 99.55 | 99.55 | 99.55 |
| IDLDMS-PTC | 99.15 | 98.84 | 99.35 |
| ODL-PTNTC | 98.73 | 97.75 | 98.40 |
| WELM Model | 97.76 | 97.67 | 97.26 |
| KELM Model | 96.66 | 97.53 | 96.69 |
| ELM Model | 96.27 | 97.27 | 96.21 |
| CNN-50x50 | 91.10 | 86.50 | 87.30 |

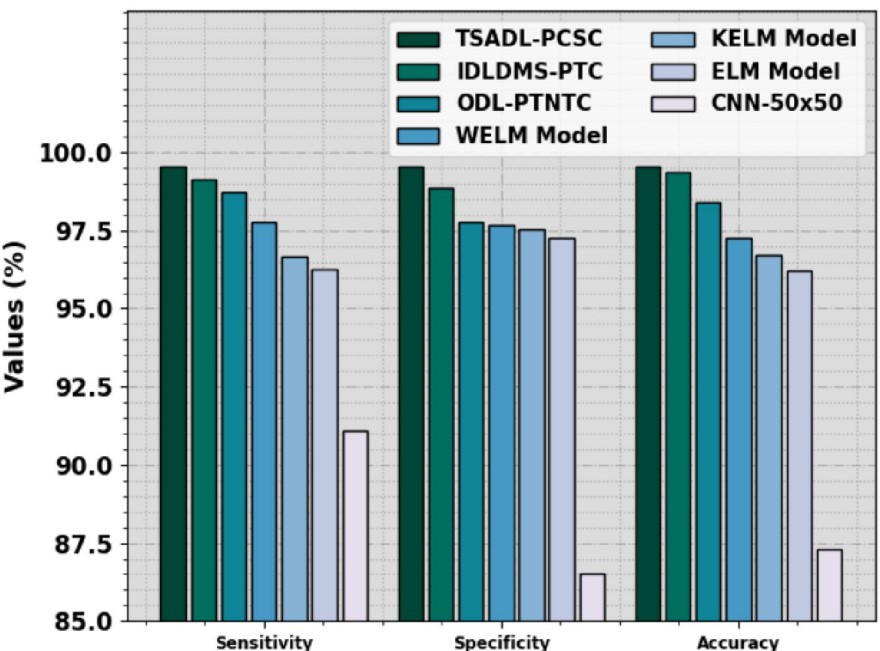

**Fig 10. Comparative outcome of TSADL-PCSC approach with other systems.**

TSADL-PCSC technique over other approaches. In the future, the performance of the TSADL-PCSC system was boosted by deep ensemble classifier algorithms.

## Author Contributions

**Conceptualization:** Hari Prasad Gandikota, Abirami S.

**Data curation:** Hari Prasad Gandikota, Abirami S., Sunil Kumar M.

**Formal analysis:** Hari Prasad Gandikota, Abirami S.

**Methodology:** Hari Prasad Gandikota, Abirami S.

**Project administration:** Hari Prasad Gandikota, Abirami S., Sunil Kumar M.

**Resources:** Hari Prasad Gandikota, Abirami S.

**Software:** Hari Prasad Gandikota, Sunil Kumar M.

**Supervision:** Hari Prasad Gandikota, Abirami S., Sunil Kumar M.

**Validation:** Hari Prasad Gandikota, Abirami S., Sunil Kumar M.

**Visualization:** Hari Prasad Gandikota, Sunil Kumar M.

**Writing – original draft:** Hari Prasad Gandikota, Abirami S.

**Writing – review & editing:** Hari Prasad Gandikota, Abirami S., Sunil Kumar M.

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
