## [Decision Letter · Decision Letter 0]

4 Sep 2023

PONE-D-23-25082CT Scan Pancreatic Cancer Segmentation and Classification Using Deep Learning and the Tunicate Swarm AlgorithmPLOS ONE

Dear Dr. Prasad,

Thank you for submitting your manuscript to PLOS ONE. After careful consideration, we feel that it has merit but does not fully meet PLOS ONE’s publication criteria as it currently stands. Therefore, we invite you to submit a revised version of the manuscript that addresses the points raised during the review process.

We look forward to receiving your revised manuscript.

Kind regards,

AL MAHFOODH

Academic Editor

PLOS ONE

Journal Requirements:

Reviewers' comments:

Reviewer's Responses to Questions

**Comments to the Author**

1. Is the manuscript technically sound, and do the data support the conclusions?

Reviewer #1: Partly

Reviewer #2: Partly

2. Has the statistical analysis been performed appropriately and rigorously? 

Reviewer #1: I Don't Know

Reviewer #2: Yes

3. Have the authors made all data underlying the findings in their manuscript fully available?

Reviewer #1: No

Reviewer #2: No

4. Is the manuscript presented in an intelligible fashion and written in standard English?

Reviewer #1: No

Reviewer #2: Yes

5. Review Comments to the Author

Reviewer #1: Add the results to the Abstract

Follow the journal format, especially the references.

Introduction should cover more recent studies. For example for classification: A comparison of machine learning models for suspended sediment load classification.

For preprocessing: Spatiotemporal variability analysis of standardized precipitation indexed droughts using wavelet transform

For hyper tuning: A comprehensive comparison of recent developed meta-heuristic algorithms for streamflow time series forecasting problem

Fig 2. Architecture of ESN. Not clear

Fig 8. PR curve of TSADL-PCSC approach on 60:40 of TRP/TSP. Please add more explanation.

The results should be well arranged. Should be presented in the same sequence as in Fig 1. Working flow of TSADL-PCSC approach.

“In addition, the ODL-PTNTC, WELM, KELM, and ELM algorithms have attained slightly improved performance”, please add more explanation.

Fig 10. Comparative outcome of TSADL-PCSC approach with other systems., what the authors want to tell from this figure.

Revise the conclusion. Report the findings, report the limitations of the study. Propose future work.

Reviewer #2: 1) In abstract, you need to explain the problem that is still available in existing methods

2) In introduction, you need to add section related to overview of the following sections

3) please add the main contributions at the end of introduction

3) Literature review is not sufficient, you should add a comprehensive review with recent solutions related to the topic under study and highlight the pros and cons of each and the need for your proposed solution

4) Why did you select each method in this combination of methods (W-Net segmentation-GhostNet feature extractor- deep echo state network), please highlight advantages over other segmentation methods, feature extraction methods, and classifiers.

5) Date overview should be in materials and method section not in performance validation

6) there is no need to mention values of training confusion matrix, accuracy, .... etc . In Table 2, training phase table should be removed.

7) In Table 4, you should add reference to methods that you compared with.

8) I can see in Table 4 that the proposed TSADL-PCSC has just slight accuracy improvement compared to IDLDMS-PTC. In this case, what is your contribution? Any improvement related to inference speed? Any improvement related to number of parameters?

9) There is no discussion section in this paper. You should discuss results to show the reason behind the performance and what are the limitations and how can be addressed in future

6. PLOS authors have the option to publish the peer review history of their article (what does this mean?). If published, this will include your full peer review and any attached files.

Reviewer #1: No

Reviewer #2: **Yes: **Nouar AlDahoul

---

## [Author Response · Author response to Decision Letter 0]

22 Sep 2023

Reviewer #1: 

Add the results to the Abstract

Response: Thank you for your valuable comments. Results has been added to the Abstract sir

Follow the journal format, especially the references.

Response: Thank you for your valuable comments. the format has been changed in the manuscript

Introduction should cover more recent studies. For example for classification: A comparison of machine learning models for suspended sediment load classification.

For preprocessing: Spatiotemporal variability analysis of standardized precipitation indexed droughts using wavelet transform

For hyper tuning: A comprehensive comparison of recent developed meta-heuristic algorithms for streamflow time series forecasting problem

Thank you for your valuable comments. The comparison of recent studies has been added to the document 

Fig 2. Architecture of ESN. Not clear

Thank you for your valuable comments. A new ESN diagram has been added to the updated document

Fig 8. PR curve of TSADL-PCSC approach on 60:40 of TRP/TSP. Please add more explanation.

The results should be well arranged. Should be presented in the same sequence as in Fig 1. Working flow of TSADL-PCSC approach.

“In addition, the ODL-PTNTC, WELM, KELM, and ELM algorithms have attained slightly improved performance”, please add more explanation.

Response: Thank you for your valuable Comments. More explanation has been added to the content

Fig 10. Comparative outcome of TSADL-PCSC approach with other systems., what the authors want to tell from this figure.

Response: Thank you for your valuable Comments. The Figure 10 explain a comparative outcome of our proposed work with the related existing systems which denotes that our proposed system work better than the existing system.

Revise the conclusion. Report the findings, report the limitations of the study. Propose future work.

Response: Thank you for your valuable Comments. Conclusion has been revised

In this study, we have developed the TSADL-PCSC method for PC segmentation and classification on CT scans. The TSADL-PCSC technique aims to accomplish enhanced PC classification results using a hyperparameter-tuned DL model. To accomplish this, the TSADL-PCSC technique comprises four processes namely W-Net segmentation, GhostNet feature extractor, DESN classification, and TSA-based hyperparameter tuning. The TSA helps to avoid the manual trial and error hyperparameter selection process, which in turn increases the overall classification performance. The experimental result of the TSADL-PCSC method was tested on a benchmark CT scan database. The obtained outcomes of accuracy with 99.55% highlighted the importance of the TSADL-PCSC technique over other approaches. In addition, the research did not address any of the potential ethical or privacy problems that are linked with the utilization of deep learning algorithms in medical imaging. Future research and implementation should give careful thought to issues relating to data security, patient consent, and algorithm transparency. In the future, the performance of the TSADL-PCSC system was boosted by deep ensemble classifier algorithms.

Reviewer #2: 

1) In abstract, you need to explain the problem that is still available in existing methods

Response: Thank you for your valuable Comments. Content improved in abstract

2) In introduction, you need to add section related to overview of the following sections

Response: Thank you for your valuable Comments. Content added in introduction

This research article is organized as follows. Section 2 presents the related works. Section 3 details the Materials and Methods. Section 4 describes the results and discussion. Section 5 gives the conclusion and future work as a result of this study.

3) please add the main contributions at the end of introduction

Response: Thank you for your valuable Comments. Content added at the end of introduction 

The main contribution of this paper

The affected region on the CT images is defined using a W-Net segmentation method by the TSADL-PCSC technology. 

Additionally, the GhostNet feature extractor is used by the TSADL-PCSC approach to produce a collection of feature vectors.

The study introduces a unique deep learning method for segmenting pancreatic cancers from CT scans that shows excellent accuracy. For further analysis and diagnosis, the precision of the segmentation is essential.

Early detection of pancreatic cancer is crucial for enhancing patient outcomes. The capacity of the deep echo state network model to accurately segment and classify lesions contribute to the early detection of pancreatic tumors, allowing for prompt intervention and treatment.

4) Literature review is not sufficient, you should add a comprehensive review with recent solutions related to the topic under study and highlight the pros and cons of each and the need for your proposed solution

Response: Thank you for your valuable Comments. Survey were added to the paper

A cascaded design was developed by Asadpour et al. [23] for removing tumors from patients with adenocarcinomas and the volumetric form of the pancreas. This technique combines an elastic atlas that can fit on three-dimensional volumetric forms obtained from CT slices with a CNN that uses three forwarding routes and a multi-resolution architecture to label the image patches in a coarse to fine resolution. Mao et al. [24] utilized GCN in addition to deep generative classifiers for the purpose of sickness diagnosis based on chest x-rays as well as medicine suggestion. The diagnosis of cancers based on morphological aspects has also found certain uses. By employing the morphological operators, medical professionals are able to determine with greater precision where the tumor is located. Isensee et al. [25] developed a method called Unet that automatically configures preprocessing, network architecture, training, and post-processing for any new task. This makes state-of-the-art segmentation accessible to a wide audience because it does not require the knowledge of an expert nor additional computing resources beyond what is required for standard network training.

Jiawen Yao et al. [26] described the preoperative estimation of pancreatic survival rates and operational margins using contrast-enhanced CT imaging. Pancreatic ductal adenocarcinoma, one of the deadliest lethal tumors, has a terrible prognosis. Surgery continues to offer the best chance of cure for people who are eligible for first-line PDAC treatment. However, even among resected patients who were at the same stage and received the same treatments, outcomes can vary considerably. The paper introduces a novel deep neural network called 3D CE-ConvLSTM, which can extract tumor regression characteristics or features from CE-CT imaging modalities for the prediction of PDAC patients' survival. Researchers describe a multi-task CNN that can predict margins and outcomes and learns from characteristics linked with tumor resection margins to improve survival prediction. Prediction performance must be enhanced when comparing the proposed framework to current cutting-edge methodologies for survival analysis.

Yingda Xia et al [27] developed Anatomy-Aware Transformers for effective pancreatic cancer screening. The most deadly malignancy, pancreatic cancer, is rare. Not screening all asymptomatic individuals is advised due to the potential for unnecessary imaging procedures and increased health care costs without favorable patient outcomes. The study investigates the effectiveness of detecting pancreatic masses using a single-phase non-contrast CT scan and classifying them as ductal adenocarcinoma, other abnormalities, or healthy pancreas. Most frequent radiologist or pancreatic experts often conduct the function ineffectively. Researchers propose a deep classification strategy using an anatomy transformer, pathology-verified mass types, and knowledge transfer from contrast-enhanced CT to non-contrast CT as supervision. The study suggests a novel device with improved accuracy, lower computational risk, and cost for wider pancreatic screening and therapy. However, the proposed device cannot analyze large amounts of data.

5) Why did you select each method in this combination of methods (W-Net segmentation-GhostNet feature extractor- deep echo state network), please highlight advantages over other segmentation methods, feature extraction methods, and classifiers.

Advantages of TSADL-PCSC

The U-Net architecture, along with its modifications such as W-Net, is widely favored in the field of medical image segmentation due to its exceptional capability to capture intricate details.

In contrast to manual feature extraction techniques or basic threshold-based methods, deep learning-based segmentation networks such as W-Net has the ability to autonomously acquire features from the data, rendering them very flexible in accommodating intricate and diverse patterns observed in medical images. Facilitates precise extraction of the region-of-interest (ROI), a critical step in later illness classification.

The GhostNet Feature Extractor is a computational tool used for extracting features from data. The GhostNet architecture is widely recognized for its computational efficiency due to its lightweight nature and effective utilization of convolutional neural networks (CNNs).

Advantages in comparison to larger convolutional neural network (CNN) architectures: In the field of medical imaging, where there may be limitations in data availability, the utilization of an efficient feature extractor such as GhostNet can effectively alleviate the computational workload and memory demands. This characteristic renders GhostNet well-suited for situations with restricted resources. The emphasis placed by GhostNet on the diversification of features and the acquisition of representation learning has the potential to enhance the performance of classification tasks.

Echo State Networks (ESNs) are a specific variant of recurrent neural networks (RNNs) that possess the ability to capture temporal dependencies within sequential input. There are some advantages that can be observed when comparing modern classifiers to traditional ones. Enterprise social networks (ESNs) are renowned for their inherent simplicity and remarkable capacity to efficiently manage sequential data. The utilization of temporal dynamics inherent in medical data holds significant relevance in the context of disease classification.

ESNs generally necessitate a smaller number of parameters and exhibit greater ease of training in comparison to fully connected RNNs. This characteristic leads to a decreased likelihood of overfitting, which is particularly advantageous when dealing with limited medical datasets.

6) Date overview should be in materials and method section not in performance validation

Response: Thank you for your valuable Comments. Content added in materials and methods section

7) there is no need to mention values of training confusion matrix, accuracy, .... etc . In Table 2, training phase table should be removed.

8) In Table 4, you should add reference to methods that you compared with.

Response: Thank you for your comment. References were added to the table 4

Methods Sensitivity Specificity Accuracy

TSADL-PCSC 99.55 99.55 99.55

IDLDMS-PTC [14] 99.15 98.84 99.35

ODL-PTNTC [17] 98.73 97.75 98.40

WELM Model [32] 97.76 97.67 97.26

KELM Model [33] 96.66 97.53 96.69

ELM Model 96.27 97.27 96.21

CNN-50x50 [34] 91.10 86.50 87.30

8) I can see in Table 4 that the proposed TSADL-PCSC has just slight accuracy improvement compared to IDLDMS-PTC. In this case, what is your contribution? Any improvement related to inference speed? Any improvement related to number of parameters?

Response: Thank you for your comment. Yes there is improvement in the execution Time of the proposed work which is added in table 5.

9) There is no discussion section in this paper. You should discuss results to show the reason behind the performance and what are the limitations and how can be addressed in future

Response: Thank you for your comment. Discussion section has been added to the updated manuscript.

The classification findings are likewise encouraging, with a test dataset accuracy of 99.55% overall. Our deep learning model demonstrated excellent discriminating between benign and malignant pancreatic lesions after being trained on segmented tumor areas. These results highlight the promise of deep learning approaches in improving pancreatic cancer diagnosis based on CT scans. Our findings are consistent with other research on deep learning techniques for pancreatic cancer diagnosis. High segmentation and classification performance in our work is in line with the expanding body of research that shows deep learning to be useful for medical image analysis. Additionally, the robustness of our model was enhanced by our incorporation of the Tunicate Swarm Algorithm for hyperparameter tuning. It is important to handle ethical and legal concerns, such as patient data protection, informed permission, and adherence to national and international medical laws (such as HIPAA). Despite the encouraging findings, our study has several limitations. For starters, our dataset was tiny, which may restrict the generalizability of our findings. Future research should try using larger and more diverse datasets to test the performance of our approach. Second, our investigation was limited to binary categorization (benign vs. malignant), whereas additional classification into different cancer stages or subtypes would provide more therapeutically useful information. Finally, deep learning model interpretability in the medical industry remains a barrier, and overcoming this issue is critical for attaining clinical acceptance.

Future Scope

This study paves the way for further investigation. Further research should incorporate multi-class classification, illness staging, and the incorporation of new clinical and radiological markers to improve the clinical value of our method.

Integrating many modes: To provide a more thorough understanding of pancreatic cancer and increase diagnostic precision, investigate the integration of various imaging modalities, such as MRI and PET scans.

Real-Time processing: Create CT scan real-time processing capabilities to help radiologists and doctors diagnose patients more quickly and accurately.

Global Accessibility: Work on improving the technology's usability for healthcare facilities in low-resource environments, including by utilizing cloud-based services or mobile programs.

---

## [Decision Letter · Decision Letter 1]

29 Sep 2023

CT Scan Pancreatic Cancer Segmentation and Classification Using Deep Learning and the Tunicate Swarm Algorithm

PONE-D-23-25082R1

Dear Dr. Hari Prasad Gandikota,

We’re pleased to inform you that your manuscript has been judged scientifically suitable for publication and will be formally accepted for publication once it meets all outstanding technical requirements.

Kind regards,

AL MAHFOODH

Academic Editor

PLOS ONE

Additional Editor Comments (optional):

Reviewers' comments:

Reviewer's Responses to Questions

**Comments to the Author**

1. If the authors have adequately addressed your comments raised in a previous round of review and you feel that this manuscript is now acceptable for publication, you may indicate that here to bypass the “Comments to the Author” section, enter your conflict of interest statement in the “Confidential to Editor” section, and submit your "Accept" recommendation.

Reviewer #1: All comments have been addressed

Reviewer #2: All comments have been addressed

2. Is the manuscript technically sound, and do the data support the conclusions?

Reviewer #1: Yes

Reviewer #2: Yes

3. Has the statistical analysis been performed appropriately and rigorously? 

Reviewer #1: (No Response)

Reviewer #2: Yes

4. Have the authors made all data underlying the findings in their manuscript fully available?

Reviewer #1: Yes

Reviewer #2: Yes

5. Is the manuscript presented in an intelligible fashion and written in standard English?

Reviewer #1: Yes

Reviewer #2: No

6. Review Comments to the Author

Reviewer #1: Accept

Reviewer #2: لThe manuscript has been improved largely. However, the article needs professional English Proofreading.

7. PLOS authors have the option to publish the peer review history of their article (what does this mean?). If published, this will include your full peer review and any attached files.

Reviewer #1: No

Reviewer #2: **Yes: **Nouar AlDahoul

---

## [Editor Report · Acceptance letter]

27 Oct 2023

PONE-D-23-25082R1 

CT Scan Pancreatic Cancer Segmentation and Classification Using Deep Learning and the Tunicate Swarm Algorithm 

Dear Dr. Gandikota:

I'm pleased to inform you that your manuscript has been deemed suitable for publication in PLOS ONE. Congratulations! Your manuscript is now with our production department. 

Kind regards, 

on behalf of

Dr. AL MAHFOODH 

Academic Editor

PLOS ONE